# Low-Rank Modular Reinforcement Learning via Muscle Synergy

**Heng Dong**[*]
IIIS, Tsinghua University
drdhxi@gmail.com

**Tonghan Wang**[*]
Harvard University
twang1@g.harvard.edu

**Jiayuan Liu**
IIIS, Tsinghua University
georgejiayuan@gmail.com

**Chongjie Zhang**
IIIS, Tsinghua University
chongjie@tsinghua.edu.cn

## Abstract

Modular Reinforcement Learning (RL) decentralizes the control of multi-joint robots by learning policies for each actuator. Previous work on modular RL has proven its ability to control morphologically different agents with a shared actuator policy. However, with the increase in the Degree of Freedom (DoF) of robots, training a morphology-generalizable modular controller becomes exponentially difficult. Motivated by the way the human central nervous system controls numerous muscles, we propose a Synergy-Oriented LeARning (SOLAR) framework that exploits the redundant nature of DoF in robot control. Actuators are grouped into synergies by an unsupervised learning method, and a synergy action is learned to control multiple actuators in synchrony. In this way, we achieve a low-rank control at the synergy level. We extensively evaluate our method on a variety of robot morphologies, and the results show its superior efficiency and generalizability, especially on robots with a large DoF like `Humanoid++` and UNIMALs.

## 1 Introduction

Deep reinforcement learning (RL) has contributed significantly to the sensorimotor control of both simulated [Heess et al., 2017, Zhu et al., 2020] and real-world [Levine et al., 2016, Mahmood et al., 2018] robots. Monolithic learning is a popular paradigm for learning control policies. In this paradigm, a policy inferring a joint action for all limb actuators based on a global sensory state is learned. Although monolithic learning has made impressive progress [Chen et al., 2020, Kuznetsov et al., 2020], it has two major shortcomings. First, the input and output space is large. For robots with more joints, learning controlling policies puts a heavy burden on the representational capacity of neural networks. Second, the input and output dimensions are fixed, making it inflexible to transfer the learned control policies to robots with different morphologies.

Modular reinforcement learning provides an elegant solution to these problems. In this learning paradigm [Wang et al., 2018], the control policy is decentralized [Peng et al., 2021], and each limb actuator is controlled by a local policy. Recent research efforts show that the local policies can learn high-performance and transferable control strategies by sharing parameters [Huang et al., 2020], communicating to each other by message passing [Huang et al., 2020], and adaptively paying attention to other actuators via graph neural networks [Kurin et al., 2020]. By exploiting the flexibility and generalizability provided by modularity, a modular policy can now control robots of up to thousands of morphologies [Gupta et al., 2021a].

---

[*]These authors contributed equally to this work.

36th Conference on Neural Information Processing Systems (NeurIPS 2022).

Despite the significant progress, modular reinforcement learning is still limited in terms of the complexity of morphological structures that can be controlled and struggles on robots with many joints like Humanoid [Kurin et al., 2020]. The large degree of control freedom presents a major challenge for learning control policies. A question is why humans can control hundreds of muscles with dexterity while the most advanced RL policy can only control less than fifteen actuators.

Studies on muscle synergies [d'Avella et al., 2003] may provide an answer. A human central nervous system decreases the control complexity by producing a small number of electrical signals and activating muscles in groups [Ting and McKay, 2007]. Muscle synergy is the coordination of muscles that are activated in synchrony. With muscle synergies, the human nervous system achieves low-rank control over its actuators. In this paper, we aim to use the inspiration of muscle synergies to reduce the control complexity and improve the learning performance of modular RL.

The first challenge of incorporating muscle synergies into modular RL is to discover a synergy structure that can promote policy learning. Neuroscience researchers factorize electrical signals [Saito et al., 2018, Falaki et al., 2017, Kieliba et al., 2018] to analyze the synergy structure, but policy signals are sub-optimal or even absent during reinforcement learning. We thus exploit the functional similarity and morphological context of actuators and use a clustering algorithm to identify actuators in the same synergy. The intuition is that muscles in a synergy typically serve the same functional purpose and have similar morphological contexts. We quantify the functional similarity by the influence of an actuator's action on the global value function, and the morphological structure is encoded as a distance matrix. To use the two types of information simultaneously, we adopt the affinity propagation algorithm [Frey and Dueck, 2007]. The synergy structure is updated periodically during learning to promptly reflect changes in value functions.

To exploit the discovered synergy structure, we design a synergy-aware architecture for policy learning. The major novelty here is that the policy learns action selection for each synergy, and the synergy actions are transformed linearly to get actuator actions. Since the number of synergies is typically much smaller than actuators, we actually learn a low-rank control policy where the physical actions are a linear mapping from a low-dimensional action space. Moreover, for better processing state information, the synergy-aware policy adopts a two-level transformer structure, which first aggregates information within each synergy and then processes information across synergies.

We evaluate our Synergy-Oriented LeARning (SOLAR) framework on two MuJoCo [Todorov et al., 2012] locomotion benchmarks [Huang et al., 2020, Gupta et al., 2021b] and in multi-task to zero-shot learning, single-task settings. SOLAR significantly outperforms previous state-of-the-art algorithms in terms of both sample efficiency and final performance on all tested settings, especially on robots with a large DoF like Humanoid++ [Huang et al., 2020] and UNIMALs [Gupta et al., 2021b]. Performance comparison and the visualization of learned synergy structures strongly support the effectiveness of our synergy discovery method and synergy-aware transformer-based policy learning approach. Our experimental results reveal the *low-rank* nature of multi-joint robot control signals.

## 2 Background

**Modular RL**. Modular Reinforcement Learning decentralizes the control of multi-joint robots by learning policies for each actuator. Each joint has its controlling policy and they coordinate with each other via various message passing schemes. Modular RL usually needs to deal with agents with different morphologies. To do so, Wang et al. [2018] and Pathak et al. [2019] represent the robot's morphology as a graph and use GNNs as policy and message passing networks. Huang et al. [2020] uses both bottom-up and top-down message passing scheme through the links between joints for coordinating. All of these GNN-like works show the benefits of modular policies over a monolithic policy in tasks tackling different morphologies. However, recently, Kurin et al. [2020] validated a hypothesis that any benefit GNNs can extract from morphological structures is outweighed by the difficulty of message passing across multiple hops. They further propose a transformer-based method, AMORPHEUS, that utilizes self-attention mechanisms as a message passing approach. AMORPHEUS outperforms prior works and our work is based on AMORPHEUS. Previous works mainly focused on effective message passing schemes, while our work aims at reducing learning complexities when the DoF of the robot is large.

**Muscle Synergy**. How the human central nervous system (CNS) coordinates the activation of a large number of muscles during movement is still an open question. According to numerous studies,

the CNS activates muscles in groups to decrease the complexity required to control each individual muscle [d'Avella et al., 2003, Ting and McKay, 2007]. According to muscle synergy theory, the CNS produces a small number of signals. The combinations of these signals are distributed to the muscles [Wojtara et al., 2014]. Muscle synergy is the term for the coordination of muscles that activate at the same time [Ferrante et al., 2016]. A synergy can include multiple muscles, and a muscle can belong to multiple synergies. Synergies produce complicated activation patterns for a set of muscles during the performance of a task, which is commonly measured using electromyography (EMG) [Tresch et al., 2002, Singh et al., 2018]. EMG signals are typically recorded as a matrix with a column for activation signals for a moment and a row for activation of a muscle [Rabbi et al., 2020]. Factorisation methods on the matrix are used to extract muscle synergies from muscle activation patterns. Four most commonly used factorization methods are non-negative matrix factorisation [Steele et al., 2015, Schwartz et al., 2016, Lee and Seung, 1999, Rozumalski et al., 2017, Shuman et al., 2016, Saito et al., 2018] , principal component analysis [Ting and Macpherson, 2005, Ting et al., 2015, Danion and Latash, 2010, Falaki et al., 2017], independent component analysis [Hyvärinen and Oja, 2000, Hart and Giszter, 2013], and factor analysis [Kieliba et al., 2018, Saito et al., 2015].

In the field of robot control, only a few works [Palli et al., 2014, Wimböck et al., 2011, Ficuciello et al., 2016] have exploited the idea of muscle synergy for dimensionality reduction to simplify the control. However, these works usually first use motion dataset from humans to obtain the synergy space and then learn to control in this synergy space. In contrast, our work learns the synergy space simultaneously with the control policy in the synergy space.

**Affinity propagation** [Frey and Dueck, 2007] is a clustering algorithm based on multi-round message passing between input data points. It does not need to pre-define the number of clusters and proceeds by finding each instance an exemplar. Data points that choose the same exemplar belongs to the same cluster.

Suppose $\{x_i\}_{i=1}^n$ is a set of data points. Define $S \in \mathbb{R}^{n \times n}$ as a similarity matrix. When $i \neq j$, the element $s_{i,j}$ at $i$th row and $j$th column is the similarity between $x_i$ and $x_j$, which can be measured as, for example, the negative squared distance of two data points. When $i = j$, the element $s_{i,j}$ represents how likely the corresponding instance is to become an exemplar. The vector of diagonal elements, $(s_{11}, s_{22}, \ldots, s_{nn})$, is called *preference*. Non-diagonal elements in $S$ constitute the *affinity* matrix. The algorithm takes $S$ as input and proceeds by updating two matrices: the responsibility matrix $R$ whose values $r_{i,j}$ represent whether $x_j$ is well-suited to be the exemplar for $x_i$; the availability matrix $A$ whose values $a_{i,j}$ quantify the appropriateness for $x_i$ picking $x_j$ as its exemplar [Frey and Dueck, 2007]. These two matrices are initialized to be zeroes and can be regarded as log-probability tables. The algorithm then alternatives between two message-passing steps. First, the responsibility matrix is updated:

$$r_{i,j} \leftarrow s_{i,j} - \max_{j' \neq j} \left(a_{i,j'} + s_{i,j'}\right). \tag{1}$$

Then, the availability matrix is updated:

$$a_{i,j} \leftarrow \min\left(0, r_{j,j} + \sum_{i' \notin \{i,j\}} \max(0, r_{i',j})\right) \text{ for } i \neq j; \quad a_{j,j} \leftarrow \sum_{i' \neq j} \max(0, r_{i',j}). \tag{2}$$

Messages are passed until the clusters stabilize or the pre-determined number of iterations is reached. Then the exemplar of $i$ is $\arg\max_j r_{i,j} + a_{i,j}$.

## 3 Method

In this section, we present our Synergy-Oriented LeARning (SOLAR) scheme that incorporates the muscle synergy mechanism into modular reinforcement learning to reduce its learning complexity.

Our method has two major components. The first one is an unsupervised learning module that utilizes the morphological structure and value information to discover the synergy hierarchy. The second is a novel attention-based policy architecture that supports synergy-aware learning. Both of the components are specially designed to enable the control of robots with different morphologies. We first introduce the problem settings and then describe the details of the two components.

**Problem settings.** We consider $N$ robots, each with a unique morphology. Agent $n$ contains $K_n$ limb actuators that are connected together to constitute its overall morphological structure. Exam-

ples of such robots that are studied in this paper include `Humanoid++` and `UNIMALs`. At each discrete timestep $t$, actuator $k \in \{1, 2, \ldots, K_n\}$ of a robot $n \in \{1, 2, \ldots, N\}$ receives a local sensory state $s_t^{n,k}$ as input and outputs individual torque values $a_t^{n,k}$ for the corresponding actuator. Then the robot $n$ executes the joint action $\boldsymbol{a}_t^n = \{a_t^{n,k}\}_{k=1}^{K_n}$ at time $t$, after which the environment returns the next state $\boldsymbol{s}_{t+1}^n = \{s_{t+1}^{n,k}\}_{k=1}^{K_n}$ corresponding to all limbs of the agent $n$ and a collective reward for the whole morphology $r_t^n(\boldsymbol{s}_t^n, \boldsymbol{a}_t^n)$. We learn a policy $\pi_\theta$ to generate actions based on states. The learning objective of the policy is to maximize the expected return on all the tasks:

$$\mathcal{J}(\theta) = \mathbb{E}_\theta \sum_{n=1}^{N} \sum_{t=0}^{\infty} \left[ \gamma^t r_t^n(\boldsymbol{s}_t^n, \boldsymbol{a}_t^n) \right], \tag{3}$$

where $\gamma$ is a discount factor. We adopt an actor-critic framework for policy learning. The critic is shared among all tasks and estimates the Q-function for each robot $n$:

$$Q^{\pi_\theta}(\boldsymbol{s}^n, \boldsymbol{a}^n) = \mathbb{E}_\theta \sum_{t=0}^{\infty} \left[ \gamma^t r_t^n(\boldsymbol{s}_t^n, \boldsymbol{a}_t^n) | \boldsymbol{s}_0^n = \boldsymbol{s}^n, \boldsymbol{a}_0^n = \boldsymbol{a}^n \right]. \tag{4}$$

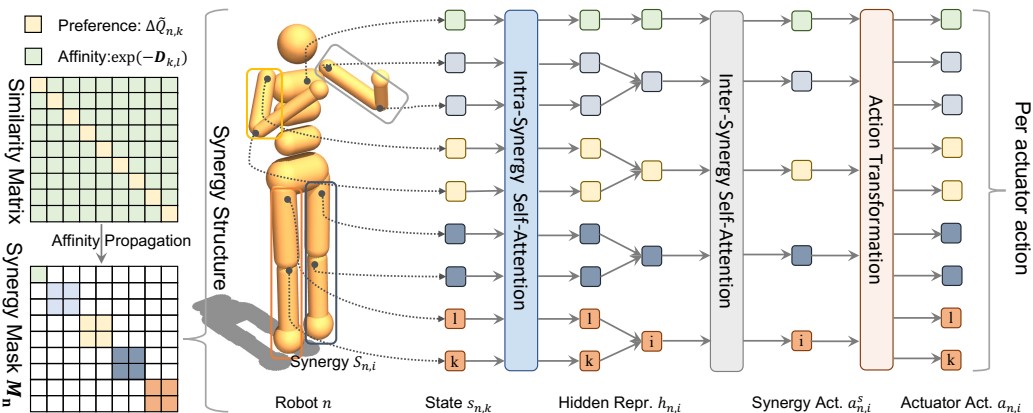

Figure 1: Synergy-aware policy learning. The intra-synergy attention module aggregates actuator information within each synergy. The inter-synergy attention module synthesizes information from all synergies to produce synergy actions. Synergy actions are then transformed linearly to obtain actuator actions. Actuator actions are of a lower rank, reducing the control complexity.

## 3.1 Discovering synergy structure

In Neuroscience, the muscle synergies are usually discovered by a factorization of electrical muscle signals during performing tasks [Todorov and Ghahramani, 2004, Rabbi et al., 2020]. Factorization is a method in hindsight – it statistically analyzes the optimal control policies of animals embodied in the electrical signals. By contrast, in reinforcement learning, we do not have the optimal control policies in advance. The synergy hierarchy learned from non-optimal policies is more likely incorrect, which will hamper policy learning. Therefore, we propose learning the synergy hierarchy by an unsupervised learning method that incorporates morphological information besides learning information. In this section, we describe our synergy hierarchy discovery methods.

Intuitively, actuators in the same synergy are activated simultaneously and they together finish a motion of an end effector. This gives us a hint that actuators with similar functions are supposed to be in a synergy. Formally, the function of an actuator (the $k$th actuator in robot $n$) can be modelled by its influence to the value function:

$$\Delta Q_{n,k} = \mathbb{E}_{\boldsymbol{s}^n, a^{n,k}, \boldsymbol{a}^{n,-k}} \left[ Q^\pi(\boldsymbol{s}^n, [a^{n,k}, \boldsymbol{a}^{n,-k}]) - Q^\pi(\boldsymbol{s}^n, [b^{n,k}, \boldsymbol{a}^{n,-k}]) \right], \tag{5}$$

where $[\cdot, \cdot]$ combines two terms, $a^{n,k}$ is the actual action of actuator $k$, $b^{n,k}$ is a default action of actuator $k$ ($b^{n,k} = 0$ in MuJoCo environments) and $\boldsymbol{a}^{n,-k}$ is the actions of actuators on robot $n$

except for the $k$th one. In practice, we use a SoftMax function to regularize $\Delta Q_{n,k}$: $\Delta \tilde{Q}_{n,k} = \exp(\Delta Q_{n,k}) / \sum_j \exp(\Delta Q_{n,j})$.

Ideally, $\Delta \tilde{Q}_{n,k}$ contains actuator clustering information, but the Q-values may be inaccurate during learning. We propose to treat $(\Delta \tilde{Q}_{n,1}, \Delta \tilde{Q}_{n,2}, \cdots, \Delta \tilde{Q}_{n,K_n})$ as the preference vector of the affinity propagation clustering algorithm (Sec. 2), i.e, the diagonal elements of similarity matrix, and in the next paragraph we further encode stable morphological information in the affinity matrix of affinity propagation, i.e, the non-diagonal elements of similarity matrix. The advantage of this design is that (1) we can simultaneously consider functional similarity and morphological context when using clustering to discover synergies and (2) static morphological information may help stabilize the clustering results.

We now discuss how to encode morphological information in the affinity matrix. A robot morphology is usually organized in a tree structure with its torso as the root node [Huang et al., 2020]. We define the adjacency matrix $\boldsymbol{A}_d \in \{0, 1\}^{K_n \times K_n}$ to represent robot's morphology, where $A_{i,j} = 1$ if actuators $i, j$ are directly linked. To measure the connectiveness of every actuator pair, we use Floyd-Warshall algorithm [Cormen et al., 2022] to compute the shortest distances between actuators based on $\boldsymbol{A}_d$, resulting in a distance matrix $\boldsymbol{D} \in \mathbb{N}^{K_n \times K_n}$. The matrix $\exp(-\boldsymbol{D})$ will serve as the affinity matrix $\boldsymbol{A}$ of affinity propagation. After defining the preference and affinity for affinity propagation, the clustering algorithms run $n_{max}$ rounds of message passing to determine actuator clusters. We treat each cluster as a synergy.

In practice, we update the synergy structure periodically during the learning process. While $\boldsymbol{D}$ remains unchanged, $\Delta \tilde{Q}_{n,k}$ changes as the Q-function is updated. Each time we run the affinity propagation with the new preference vector and then freeze the synergy structure for a while.

## 3.2 Learning synergy-aware low-rank policies

We now describe how we design a synergy-aware actor-critic learning framework that exploits the synergy hierarchy discovered in Sec. 3.1.

The core idea is that the policy generates control signals (actions) for synergies instead of actuators. As the number of synergies is much fewer than actuators, the learning complexity is reduced. The synergy actions will be combined linearly to generate physical actions for actuators. In this way, we learn a low-rank control policy – the physical actions are actually a linear combination of actions in a low-dimensional space.

Suppose that we find $L_n$ synergies $\{S_{n,i}\}_{i=1}^{L_n}$ for the $n$th robot. We use a mask $\boldsymbol{M}_n \in \{0, 1\}^{K_n \times K_n}$ to represent the synergy structure. The element at the $i$th row, $j$th column of $\boldsymbol{M}_n$ is 1 if actuator $i$ and $j$ are in the same synergy. The policy $\pi_\theta$ consists of three components: an intra-synergy attention module, an inter-synergy attention module, and an action transformation matrix as in Figure 1.

The first module aggregates information of actuators in a synergy. Input to this module is the joint state $s_n$, and this module outputs a hidden representation $h_{n,i} \in \mathbb{R}^d$ for each synergy $S_{n,i}$. In practice, we use a transformer with a two-head self-attention layer where the attentions between actuators belonging to different synergies are masked out. The output of the self-attention layer is of the size $K_n \times d$. We then use mean pooling to aggregate information of actuators in the same synergy. In this way, we get a hidden state $h_n \in \mathbb{R}^{L_n \times d}$, with each row being the aggregated information of a synergy.

The second module aggregates information from different synergies and outputs the synergy actions. It has two multi-head self-attention layers. We feed $h_n$ into this these layers and get vector-valued synergy actions $\boldsymbol{a}_n^s \in \mathbb{R}^{L_n \times 1}$. The synergy actions are then transformed by learnable matrices $\boldsymbol{T} \in \mathbb{R}^{K_n \times L_n}$, and we obtain actions for actuators by $\boldsymbol{a}_n = \boldsymbol{T} \boldsymbol{a}_n^s$. One question about the transformation matrix $\boldsymbol{T}$ is that $L_n$ may change during the learning process, requiring the dimension of $\boldsymbol{T}$ to be dynamic. In Appendix A, we discuss how to deal with this problem.

The critic uses the same intra- and inter-synergy attention structure as the policy for processing state information. The outputs of the inter-synergy attention module are fed into a fully-connected network to obtain the Q-value estimation. In practice, we use TD3 [Fujimoto et al., 2018] for the training of the policy and the critic. The intra- and inter-synergy transformer and the transformation

matrix are updated in an end-to-end manner. We provide more details of our learning framework in Appendix A.

# 4 Experiments

In this section, we benchmark our method SOLAR on various MuJoCo [Todorov et al., 2012] locomotion tasks. we evaluate the effectiveness of SOLAR by asking the following questions: (1) Can our SOLAR outperforms other modular RL approaches when simultaneously trained on a large number of diverse agents with different morphologies? (Sec. 4.2) (2) Can the learned policy in multi-task environments generalize to new tasks with unseen morphologies? (Sec. 4.3) (3) How does SOLAR learn synergy clusters and how do synergy clusters facilitate learning? (Sec. 4.4) (4) Can SOLAR scale to single-tasks with numerous joints? (Sec. 4.5). For qualitative results, please refer to the videos on our project website[*]. And our code is available at GitHub[*].

## 4.1 Experiment setup

We run experiments on two benchmarks, and report several results in this section. For additional results, please refer to Appendix B. We test all methods with 4 random seeds and show the mean performance as well as 95% confidence intervals.

**Environments**. For multi-task and zero-shot evaluation, we adopt the widely-used modular MTRL benchmarks [Huang et al., 2020, Kurin et al., 2020, Hong et al., 2021], which are created based on Gym MuJoCo locomotion tasks by Huang et al. [2020]. In this section, we report results on 3 in-domain settings: (1) 6 variants of walker [`Walker++`] (2) 8 variants of humanoid [`Humanoid++`] (3) 3 variants of hopper [`Hopper++`]; and 1 cross-domain settings: all 6 variants of walker, all 8 variants of humanoids, and all 3 variants of hopper [`Walker-Humanoid-Hopper++`]. For single-task evaluation, we sample 3 morphologies from UNIMALS [Gupta et al., 2021b,a], which are obtained from evolution and have numerous joints, making it suitable to evaluate the performance of modular RL policies in single-tasks. For a detailed description of the environments please refer to Gupta et al. [2021b] and Appendix A.

**Baselines**. In this section, we compare our method SOLAR against state-of-the-art modular RL methods SMP [Huang et al., 2020] and AMORPHEUS [Kurin et al., 2020]. SMP passes messages along the limbs using a bottom-up and top-down scheme. SOLAR and AMORPHEUS use transformer-based actors and critics, whose message passing schemes are realized by self-attention. We also compare SOLAR with standard TD3-based non-modular RL: Monolithic. Please refer to Appendix A for more details about baselines.

**Ablations**. There are two contributions that characterize our method. (1) An unsupervised learning module that utilizes the value information and the morphological contextual structure to discover the synergy hierarchy. (2) A novel attention-based synergy-aware policy architecture that supports synergy-aware learning. Our novelties are mainly about the discovery and utilization of synergies, and these two contributions closely rely on each other. If we totally ablate synergies, we will get AMORPHEUS. Therefore, the effectiveness of synergy-based learning can be demonstrated by comparing SOLAR against AMORPHEUS. We further design the following ablations: (1) AMORPHEUS *w/ synergy mask*. Incorporate synergy-aware inter-cluster masks into AMORPHEUS to test the contribution of synergy hierarchy. (2) SOLAR *w/o preference*. Remove value information (the preference vector) from affinity propagation of SOLAR. In this case, affinity propagation is based solely on morphological information, and we can test the effects of value information.

**Implementations**. We use TD3 [Fujimoto et al., 2018] as the underlying reinforcement learning algorithm for training the policy over all baselines, ablations and our method for fairness. We implement SOLAR in the AMORPHEUS codebase. And like AMORPHEUS, there is no weight sharing between actor and critic. SOLAR uses a traversal-based embedding for action transformation, which may incorporate structural information. For a fair comparison, we concatenate these embeddings into the original observation vectors of each environment which is applied to all tested algorithms.

---

[*]`https://sites.google.com/view/synergy-rl`
[*]`https://github.com/drdh/Synergy-RL`

## 4.2 Multi-task with different morphologies

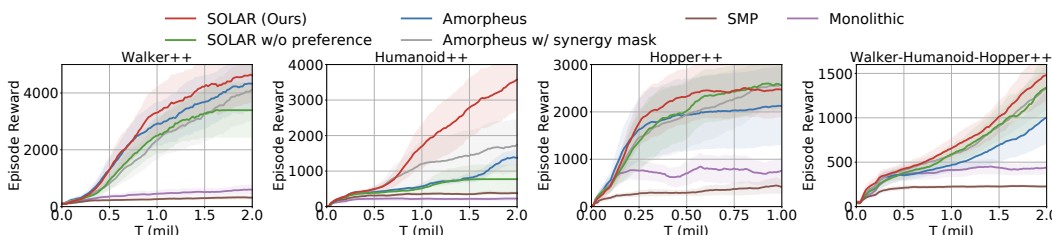

Figure 2: Multi-task performance of our method SOLAR compared to baselines and ablations.

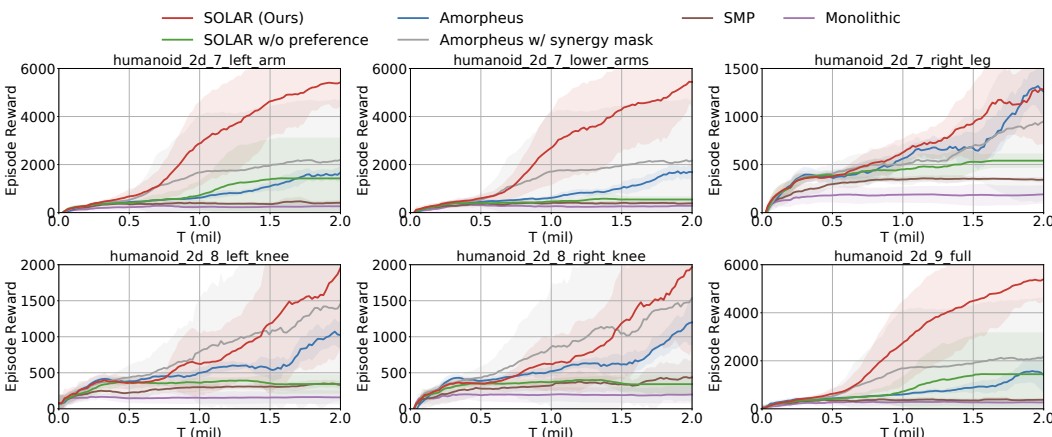

Figure 3: Multi-task performance of our method SOLAR compared to baseline and ablations on `Humanoid++`.

We summarize the multi-task results in Figure 2. SOLAR outperforms the previous state-of-the-art algorithms AMORPHEUS and SMP in all settings. Furthermore, the performance gap between SOLAR and AMORPHEUS is notably larger in `Humanoid++` which has the maximum number of joints among the tested settings. We also plot the learning curves of six training variants (Figure 3) and two testing variants (Figure 4) of `Humanoid++`. These results consistently demonstrate the effectiveness of SOLAR in training robots with a large degree of freedom (DoF). We speculate that the reason of the effectiveness is that SOLAR controls robots in a lower-rank action space than the original space and the action spaces of robots with a large DoF are more likely to be reduced via the synergy mechanism. To further validate this speculation, we conduct extra evaluations on robots with much more joints in Sec. 4.5.

As for ablations, AMORPHEUS *w/ synergy mask* outperforms AMORPHEUS in most tasks in Figure 2 and Figure 3, which indicates the effectiveness of the unsupervised learning method that discovers synergy clusters and synergy-aware masks in the policy. However, SOLAR *w/o preference* performs comparable to or even worse than AMORPHEUS, which reflects the prominent usefulness of value information.

## 4.3 Zero-shot generalization

In this section, we benchmark SOLAR in a zero-shot learning setting where the policy is trained in multiple tasks and then generalized to unseen tasks. We compare SOLAR with AMORPHEUS, SMP, AMORPHEUS *w/ synergy mask* and SOLAR *w/o preference* in `Humanoid++`, which has the largest number of joints among the tested environments. `Humanoid++` has eight variants of humanoids, where six of them are used as training tasks (see Figure 3) and the other two are used as testing tasks (Figure 4). To obtain the results of testing tasks at each time step, we periodically load policy network parameters when training and evaluate the policy in these tasks. The trajectories on testing tasks will be discarded and will not be used for learning.

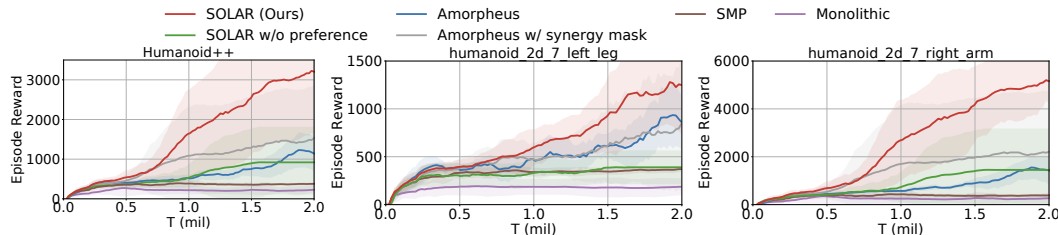

Figure 4: Zero-shot performance of our method SOLAR compared against baseline and ablations.

As shown in Figure 4, SOLAR outperforms all the baselines by a large margin in these unseen tasks. SOLAR learns faster than other baselines and shows higher sample efficiency in Figure 3, showing that the knowledge obtained from training tasks is effectively used in unseen tasks.

## 4.4 Analysis of synergies

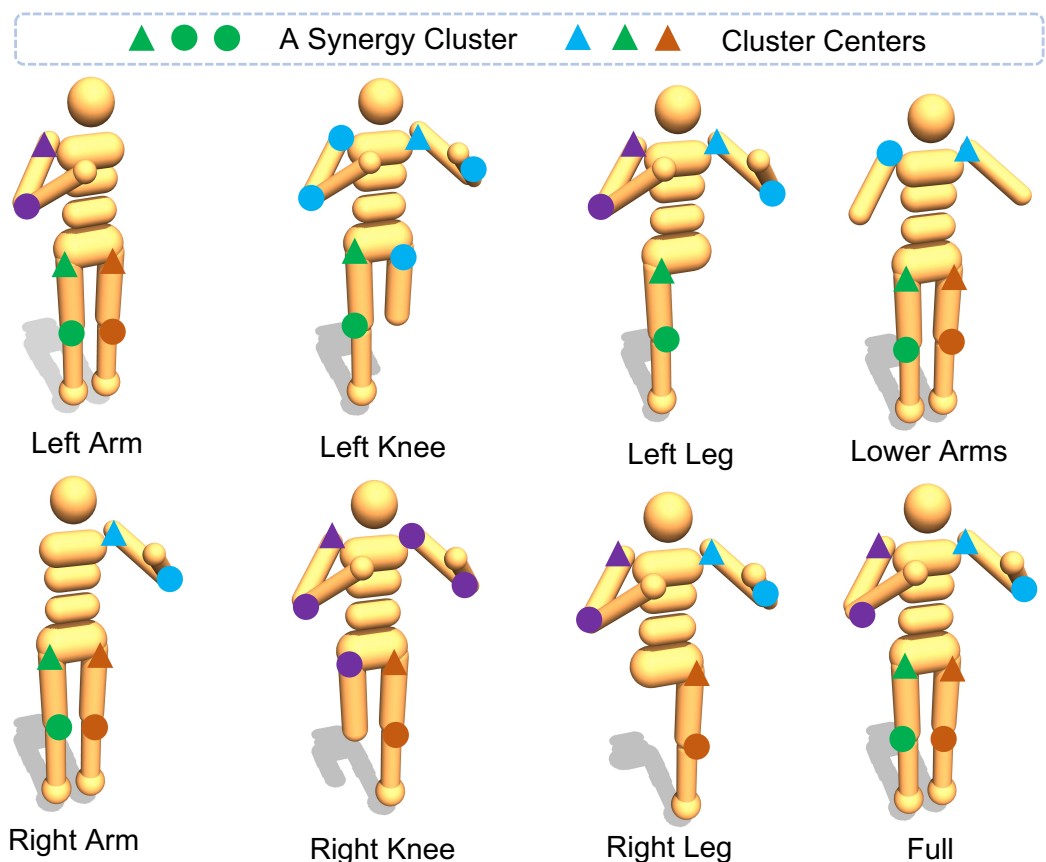

Figure 5: Synergy clustering results of SOLAR in `Humanoid++`. Different colors represent different synergy clusters, and joints marked with the same color are in the same cluster. Joints marked with triangles are the centers of their corresponding clusters.

To investigate why SOLAR performs better than baselines, in this section, we visualize the synergy clusters learned by SOLAR and the evolutionary process of synergy clusters in `Humanoid++`.

Affinity propagation which we used as the method to discover synergy clusters can naturally outputs clustering results and the cluster centers in each cluster. We first visualize the synergy clusters of SOLAR in `Humanoid++` at the end of training in Figure 5. Here we use different colors and two shapes to mark each joint in robots. Different colors represent different synergy clusters, and joints

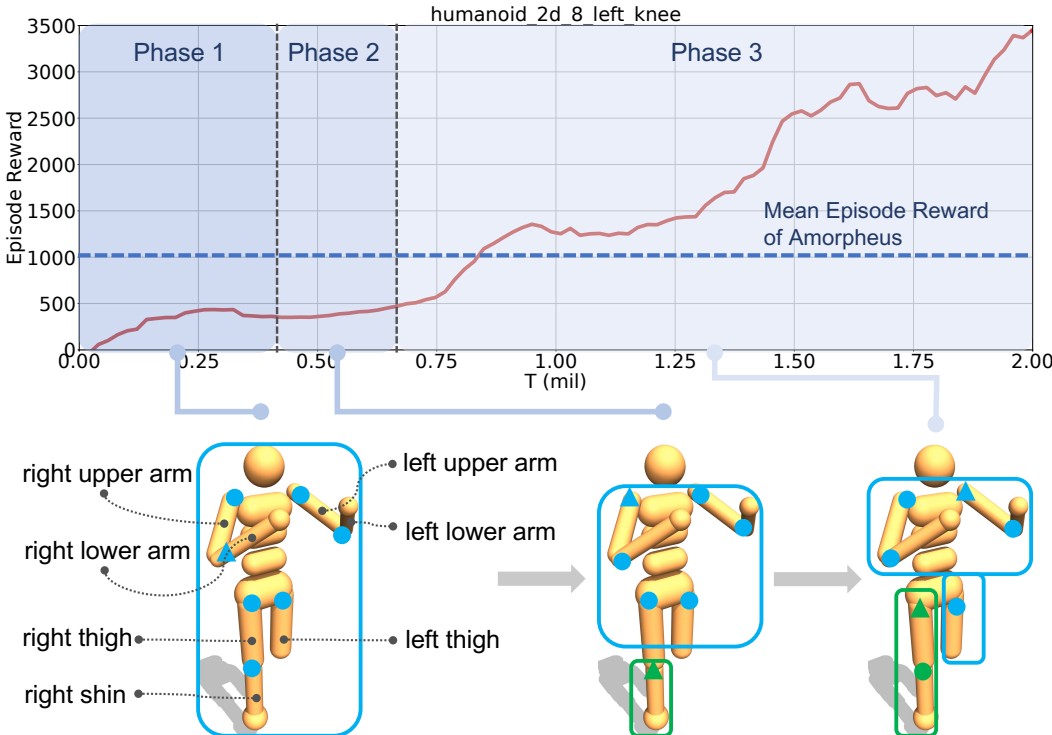

Figure 6: Synergy structure evolution of SOLAR in Humanoid. Phase are divided according to the change of synergy clusters and synergy clusters are masked with colored shapes.

marked with the same color are in the same cluster. Joints marked with triangles are the centers of their corresponding clusters. For example, the *Full humanoid* in the lowermost right corner has 4 synergy clusters, and the joints between the torso and two thighs and two arms are the cluster centers.

To further investigate how these clusters evolve, we visualize the learning process of *Left Knee Humanoid* in Figure 6. According to the change of synergy clusters, the learning process can be divided into three phases. In Phase 1, the episode reward is very low and $\Delta\tilde{Q}$ may only provide noisy information, which results in all joints being clustered in a same cluster. In Phase 2, SOLAR finds that the right shin is important and divides it into a separate cluster. Then in Phase 3, SOLAR finally obtains a suitable clustering solution and the episode reward increases correspondingly.

The visualization results reflect two facts: (1) close joints are more likely to be in the same synergy cluster, and (2) joints near the torso may be more influential than those who are far from the torso, and are thus selected as the cluster centers. These clusters can reduce learning and controlling complexities but are hard to discover through a standard learning mechanism. Instead, SOLAR makes use of early, less accurate learning information $\Delta\tilde{Q}$ and structural information via unsupervised learning to discover synergy clusters. The synergy clusters then further facilitate learning.

### 4.5 Single-task with numerous joints

To consolidate the speculation in Sec. 4.2 that SOLAR is suitable for tasks with numerous joints, we test SOLAR in three single-robot tasks sampled from UNIMALS [Gupta et al., 2021b]. The average number of joints is about twice as many as that in `Humanoid++`. For more details about these tasks, please refer to Appendix A. We showcase the results in Figure 7. SOLAR learns faster than state-of-the-art AMORPHEUS, showing much higher sample efficiency, and again outperforms it in average episode rewards.

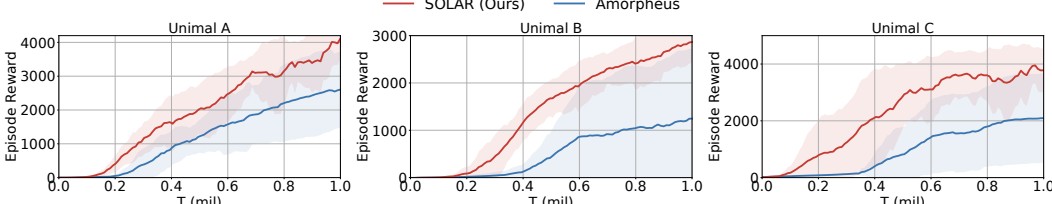

Figure 7: Multi-Task performance of our method SOLAR compared to AMORPHEUS

# 5 Conclusions

In this paper, we use the inspiration of muscle synergies from neuroscience to reduce the control complexity of modular reinforcement learning in tasks with a large number of degree of freedom. We present SOLAR that leverages both structural information and policy learning information to learn synergy clusters. Moreover, a stacked-transformer architecture is proposed for learning synergy actions which are combined linearly to produce low-rank actuator actions. The evaluations of SOLAR on various tasks and settings showcase its effectiveness. The authors do not see obvious negative societal impacts of the proposed method.

# Acknowledgments

This work is supported in part by Science and Technology Innovation 2030 New Generation Artificial Intelligence Major Project (No. 2018AAA0100904) and National Natural Science Foundation of China (62176135).

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
