# OpenReview forum: "Low-Rank Modular Reinforcement Learning via Muscle Synergy"
_NeurIPS.cc/2022/Conference — NeurIPS 2022 Accept_

### Official Review · Reviewer_tr2n · 2022-07-09

**Rating:** 6
**Confidence:** 4
**Soundness:** 3 good
**Presentation:** 3 good
**Contribution:** 3 good

**Summary:**

In this paper, the authors focus on the problem of modular RL for systems with a large degrees of freedom such as humanoid characters.

The central idea of this work can be divided into two parts. The first is the use of an unsupervised clustering algorithm to group different actuators on the agent based on their impact on the value function and the agent’s morphology. This is done using an existing algorithm named affinity propagation. The second part is an attention-based architecture that achieves a hierarchical controller where the policy produces low dimensional synergy actions, which are then mapped to per-actuator actions.

In the experiments, the method is evaluated on locomotion tasks for a variety of morphologies like walker, humanoid, unimal, etc. When compared to several state of the art baseline methods, the proposed algorithm shows superior learning performance. An ablation study was also carried out to demonstrate the importance of the proposed idea of learning synergy and the attention-based model.

**Questions:**

In addition to some of the issues mentioned above, I also have the following questions:
1. When training in a multi-task setting, do all tasks (variant of the agents) share the same critic and actor weights?
2. For synergy of different morphologies, my understanding is that they are independently computed, is that right? I.e. the subfigures in Fig 5 are independent even some of them share the same color?
3. How frequent are the actor-critic update interleaved with the update in the synergy?
4. According to the text, the motivation for the proposed synergy formulation is that the one used in neuroscience requires optimal policies and the clustering could be erroneous. However, the proposed method also suffers from this issue as value functions would also be wrong at the beginning. By interleaving the policy learning and synergy discovery this doesn't really seem to be a problem, as demonstrated in Figure 6. I'm wondering if there are other issues or potential benefits with using a more neuroscience-inspired approach.


**Limitations:**

The author didn't discuss much limitation about the proposed work. I would suggest adding some discussions in terms of what kind of problem the current formulation may not be suitable for.

**Strengths And Weaknesses:**

Strengths:
1) The idea of synergy for lowering the control complexity in high dof systems is reasonable and the proposed method for finding the synergy amongst actuators is solid.
2) The experiments in general support the proposed idea and achieves good performance in highly challenging control problems.
3) The analysis is pretty thorough, including an ablation study to show the impact of the proposed ideas, as well as a visualization of the discovered synergy throughout learning, which is helpful in gaining insights into the algorithm.
4) The method could potentially be applied to other high dof control problems.

Weaknesses:
1) The proposed seems especially effective for the humanoid task, while achieving comparable results for the baselines on the other tasks. It’d be helpful to have some discussion regarding why humanoid shows the biggest difference.
2) Currently the tasks have been focused on one locomotion task (go forward). If more tasks can be demonstrated it can further support the effectiveness of the method.
3) Some details are missing, as mentioned below.

---

> ### Author Response · Authors · 2022-08-02
> **Thanks very much for the insightful reviews! New experiments on Manipulation task and clarification of other questions**
>
> Thanks very much for the insightful reviews and suggestions. Here we provide additional experimental results and explanations for your questions. For the issues raised in the Question section, we have updated our paper to provide the suggested details.
>
> > **Weakness 1**: It’d be helpful to have some discussion regarding why humanoid shows the biggest difference.
>
> The reason is that Humanoid++ have the largest number of actuators. Humanoid++ have an average of 6.5 actuators, while they only have 2.875 synergies in Figure 5. In comparison, the average numbers of actuators and synergies are 3.0 and 2.0 for Hopper++ in Figure 10 and 3.5 and 2.0 for Walker++ in Figure 11. The difference between the number of actuators and synergies is the most significant on Humanoid++. Therefore, the benefits of low-rank control achieved by synergy-based policies are the most significant on Humanoid++.
>
> > **Weakness 2**: Currently the tasks have been focused on one locomotion task (go forward). If more tasks can be demonstrated, it can further support the effectiveness of the method.
>
> As suggested by the reviewer, we further test Manipulation task to show the effectiveness of our method in Appendix B.1. Please re-download the paper for more details. In brief, Manipulation task requires robots to first reach a box and then push the box to a randomly generated goal, which involves interaction with the box object and is much more difficult than Locomotion task. We compare our method with baseline MetaMorph, which is based on Amorpheus and uses a dynamic replay buffer balancing technique to deal with a large number of robots. We show the multi-task results in Figure 8 and the zero-shot performance in Table 4 in the Appendix. Our method outperforms MetaMorph by a large margin, which further exhibits the effectiveness of our method.
>
> > **Question 1**: When training in a multi-task setting, do all tasks (variant of the agents) share the same critic and actor weights?
>
> Yes, all variants of the robots share the same actor and critic weights. To achieve this sharing, we implemented our actor and critic network based on the transformer, which is capable of handling inputs of variable length. These results demonstrate the generalizability of our method.
>
>
> > **Question 2**: For synergy of different morphologies, my understanding is that they are independently computed.
>
> Yes, the reviewer is right. Synergies of different morphologies are computed independently. Subfigures in Figure 5 are computed independently.
>
> > **Question 3**: How frequent are the actor-critic update interleaved with the update in the synergy?
>
> We update the synergy every two updates of the policy network. Besides, as we use TD3 as the base algorithm, the policy network itself is updated every two updates of the critic network.
>
> > **Question 4**: I'm wondering if there are other issues or potential benefits with using a more neuroscience-inspired approach.
>
> Using a neuroscience-inspired approach may involve the following issues:
>
> 1. Conventional muscle synergy studies typically use factorization methods (NMF, PCA, ICA, and FA) to discover synergy structures. These methods need to pre-define the number of synergies or pre-define other variance thresholds, while our method does not need to specify them.
>
> 2. Conventional muscle synergy studies focus more on a fixed number of muscles or actuators, which can not be directly extended to robots with a variable number of actuators. Our method uses self-attention and can easily handle different robots.
>
>
> > **Limitation 1**: The author didn't discuss much limitation about the proposed work.
>
> We have added discussions about the limitations of our work in Appendix B.5.
>
> When generalizing to unseen robots with larger numbers of actuators than training robots, the embeddings of testing robots are not learned, which will hamper the zero-shot performance. One possible future direction will involve designing a more scalable actuator embedding method. Moreover, SOLAR is more suitable for robots with a large number of actuators. Our approach is able to reduce a great degree of control complexity for these robots. But for robots with few actuators, SOLAR may only have a little positive impact and the learning of synergy-aware policy may even damage the performance.

---

> > ### Comment · Reviewer_tr2n · 2022-08-09
> > **Thank you for the response**
> >
> > The authors response has addressed my questions and concerns about the work. The additional box pushing task is quite interesting to see. Thus I'd recommend acceptance of the paper.

---

### Official Review · Reviewer_NEKf · 2022-07-11

**Rating:** 7
**Confidence:** 3
**Soundness:** 3 good
**Presentation:** 2 fair
**Contribution:** 3 good

**Summary:**

In this paper, the authors proposed a new modular reinforcement learning algorithm using unsupervised learning. By auto clustering joints that have synergies, i.e. based on their contribution to the tasks through value functions, the proposed method alleviates the control to the synergy space instead of the joint space. Then, a learned mapping function will convert the synergy action to joint action signals. With the proposed framework, the authors show that they can significantly outperform the SOTA baselines SMP and AMORPHEUS in many multiple task benchmarks.

**Questions:**

See the weaknesses section for revising recommendations.

**Ethics Review Area:**

["I don’t know"]

**Strengths And Weaknesses:**

The strengths of the paper:

The results of the paper show a clear edge from the proposed framework in many benchmarks.

The authors did a good amount of ablation study. I especially like how the authors visualize the synergy group’s propagation over time in Figure 5.

The weaknesses of the paper:

The description of affinity propagation can be more clear. Currently affinity propagation is introduced in the background section and the writing is very similar to what has been written on Wikipedia. It is better to incorporate this section to methods, and define the similarity, responsibility and availability matrices in the context of the proposed system. For example, I assume that the exponential of the distance matrix in section 3.1 is used for the off-diagonal elements of the similarity matrices while delta-Q values are used as the diagonal elements. Consider re-writing this part and making it clear..

Sometimes, notations are used without proper definition. For example, the Q function in Eq (5) suddenly takes three different types of inputs, with both bold and non-bold symbols. This makes it difficult for the readers to understand. Consider revising.

No illustration diagrams of the proposed network architecture for section 3.2. It is a bit hard to decipher the data flow just from the text descriptions.

---

> ### Author Response · Authors · 2022-08-02
> **Thanks a lot for your helpful reviews! We provide clarification of the questions**
>
> Thanks for the comments and helpful suggestions. Here we provide detailed clarifications to your questions.
>
> > **Weakness 1**: The description of affinity propagation can be more clear. I assume that the exponential of the distance matrix in section 3.1 is used for the off-diagonal elements of the similarity matrices while delta-Q values are used as the diagonal elements.
>
> Thanks for the suggestion. We have rewritten the background of Affinity Propagation in Sec.2 and added extra details in Sec.3.1. The reviewer is right about how to compute the similarity matrix. In brief, we treat delta-Q values $\Delta \tilde{Q}_{n,k}$ as the diagonal elements of the similarity matrix and use the exponential of the negative distance matrix $\exp(-\mathbf{D})$ as the non-diagonal elements of the similarity matrix. Then the affinity propagation algorithm proceeds based on the similarity matrix.
>
>
> > **Weakness 2**: Some notations are used without proper definition.
>
> Thanks for the suggestion. We have revised Equation 5 to make it properly defined. The second and third inputs are now combined to become the joint action input to the Q function.
>
> > **Weakness 3**: No illustration diagrams of the proposed network architecture for section 3.2.
>
> Thanks for the suggestion. We have updated the diagram of the proposed network architecture in Figure 1 in the revised paper.

---

### Official Review · Reviewer_a1E4 · 2022-07-12

**Rating:** 7
**Confidence:** 3
**Soundness:** 3 good
**Presentation:** 3 good
**Contribution:** 3 good

**Summary:**

The authors propose a Synergy-Oriented LeARning (SOLAR) framework that exploits the redundant nature of DoF in robot control.  The method is inspired by the well-known idea of "muscle synergy" which the human central nervous system has and reduces the DOF of the whole-body musculoskeletal system. The synergy is learned in an unsupervised manner.  The experimental results show the effectiveness of the approach.

**Questions:**

To what extent does the proposed approach share the same idea with the conventional kinematic or muscle synergies which are often analyzed by PCA. It may be fruitful to show the mathematical comparison explicitly. I think many ML researchers may not know muscle synergy and researchers who are familiar with muscle synergy (e.g., people studying human body control, musculoskeletal system, CPG, and so on) don't know much about this type of ML studies. This paper potentially becomes an important paper bridging the two separated communities. From that viewpoint, the connection between the proposed method and the conventional notion of muscle synergy is better to be described more clearly in my opinion (regarding the relation to the self-attention mechanism in particular).

Do you even hypothesize that human muscle synergies are organized in this way, or do you have any implications about the human muscle synergy obtained through this study?  Such a discussion may add extra value to the paper.

**Limitations:**

It is better to compare not only modular RL methods but also SOTA central RL methods will be beneficial for discussion.

**Strengths And Weaknesses:**

+ Application of the notion of muscle synergy to modular reinforcement learning (in an unsupervised manner) is quite new and interesting.
+ Experimental results show the effectiveness of the method.
+ Emerged synergies are analyzed and exhibit interesting results.
- The relationship between the conventional muscle synergy and the self-attention mechanism is slightly unclear.

---

> ### Author Response · Authors · 2022-08-02
> **Thanks for your insightful comments! New experiments for Monolithic RL and clarification of other questions are added.**
>
> Thanks for your insightful comments and helpful suggestions. Here we provide detailed clarifications to your questions.
>
> > **Question 1**: The connection between the proposed method and the conventional notion of muscle synergy.
>
> Mathematically, conventional muscle synergy studies typically use non-negative matrix factorization (Rabbi et al, 2020) and aim to solve the following optimization problem:
>
> $\min_{H,A} \|U - HA \|_F$,
>
> where $H\in\mathbb{R}^{N\times M}, A\in\mathbb{R}^{M\times T}$ (the elements of $H,A$ are non-negative) and $U\in \mathbb{R}^{N\times T}$ is a given matrix of observed control signals. The element at $i$th row and $t$th column of $U$ is the control signal to muscle $i$ at timestep $t$. In this optimization problem, the number of synergies, $M$ (where $M<N$), is typically pre-defined or chosen according to a pre-defined reconstruction error threshold. By solving this problem, one can discover the synergy structure by observing matrix $H$. Conventional muscle synergy studies that use other factorization methods (PCA, ICA, and FA) share a similar optimization problem with different matrix constraints.
>
> We study a similar optimization problem but with additional constraints:
>
> $\max_{U,H,A} \sum_t \gamma^t R(s_t, U_t) - \|U - HA \|_F.$
>
> Here $s_t$ is the environment state at timestep $t$, and $U_t$ is the column $t$ of matrix $U$, i.e., actions at timestep $t$. And $R(s_t,U_t)$ is the reward of choosing action $U_t$ at state $s_t$.
>
> The differences are:
>
> 1. We additionally maximize the expected return of muscle actions.
>
> 2. In our formulation, the number of synergies is not pre-defined but is learned in an unsupervised manner.
>
> 3. The matrix $U$ is also not given but is generated by an attention-based policy. This policy is optimized to maximize return as well as to minimize the decomposition loss (term 2).
>
> In summary, we share a common optimization objective with conventional muscle synergy studies. However, we need to additionally learn the number of synergies and control signals which are inputs in the conventional studies. Structurally, our framework gives an attention function class that covers a synergy decomposition solution that can minimize the decomposition loss while enabling an efficient control policy.
>
> We added this discussion about conventional muscle synergy studies in Appendix B.4 of the paper.
>
>
> > **Question 2**: Do you even hypothesize that human muscle synergies are organized in this way, or do you have any implications about the human muscle synergy obtained through this study?
>
> We hypothesize that the human nervous system achieves low-rank control over its actuators via activating muscles in synchrony, and our method is inspired by this hypothesis.
>
> Our work may have the following implications for human muscle synergy:
>
> 1. Conventional muscle synergy studies discover the synergy structures via analyzing human behaviors, which does not involve how synergy structures affect behaviors. However, the synergy structure evolution in Figure 6 of our paper may indicate that synergy structure has a huge impact on behaviors or policy learning, that is, correct synergy structures are necessary for learning optimal policies.
>
> 2. Our work studies the synergy structure of robots with different actuators in Figure 2, and the performance improvements of Solar against baselines increase as the number of actuators increases. This implies that the idea of synergy is more suitable for handling high DoF issues.
>
> > **Limitation 1**: It is better to compare not only modular RL methods but also SOTA central RL methods will be beneficial for discussion.
>
> Thanks for the suggestion. We have conducted extra evaluations and added the new results in the revised paper. Please re-download the paper for more details. Here we briefly discuss the evaluation results.
>
> Following the setup by Huang et al., 2020, we choose TD3 as the standard monolithic RL baseline.  The actor and critics of TD3 are implemented by fully-connected neural networks, which take the concatenation of observations of all actuators as input. Since the number of actuators varies in different robots, the dimension of observations is incompatible. To overcome this issue, we zero-pad the observations and actions to the maximum dimension across all robots.
>
> The new results are shown in Figure 2,3,4,9 of the paper. Monolithic RL performs poorly in all multi-task environments, which indicates that monolithic RL is not suitable for these incompatible settings. But for single-task environments, Monolithic RL archives comparable performance to SOLAR.
>
> ---
>
> Mohammad Fazle Rabbi et al,. Non-negative matrix factorisation is the most appropriate method for extraction of muscle synergies in walking and running. Scientiﬁc reports, 10(1):1–11, 2020.
>
> Wenlong Huang et al,. One policy to control them all: Shared modular policies for agent-agnostic control. In International Conference on Machine Learning, pages 4455–4464. PMLR, 2020.

---

> > ### Comment · Reviewer_a1E4 · 2022-08-08
> > **Thanks for the reply**
> >
> > Thanks for the reply and the revision.
> > I think the paper is insightful not only for the AI community but also for other scientific communities related to "muscle synergy."
> > I will keep my score as it is.

---

### Author Response · Authors · 2022-08-07
**General Response to Reviewers**

We thank our respected reviewers for providing valuable feedback on the paper.  The paper has been revised with the following changes:

- New experiments on Manipulation task in Appendix B.1.
- New experiments for another baseline: Monolithic RL in Figure 2,3,4.
- Revised framework with more illustrations in Figure 1.
- Discussions of connection to conventional notion of muscle synergy in Appendix B.4.
- More discussions of limitations in Appendix B.5.
- Rewriting of affinity propagation background in Sec. 2.

Kindly let us know if our responses below addressed your concerns. If you have any further questions or comments, please post them and we will be happy to have further discussions.

---

### Meta-Review · Area_Chair_3nDY · 2022-08-27

**Recommendation:** Accept
**Confidence:** Certain

**Metareview:**

This paper propose an approach for learning low-rank synergies for morphology-generalizable robot control.

All the reviewers agree that the paper is interesting and a valuable contribution.
Hence, I recommend acceptance.

Additional comments:
- The related work section does not cover virtually any past work from robotics that deal with synergies and dimensionality reduction for large action-spaces. It would be good to include some of this literature to better place your work.
- The dimensionality of the action space in the environments used is not very high. However, the fact that in neither the manuscript nor the appendix the values of K (i.e., the number of actuators) are specified might create ambiguity. It would be good to make these values more visible.

**Award:**

No

---

### Decision · Program_Chairs · 2022-09-14

Accept